# The effect of a harmful algal bloom (*Karenia selliformis*) on the benthic invertebrate community and the sea otter (*Enhydra lutris*) diet in eastern Hokkaido

Jackson Johnstone[1]*, Ippei Suzuki[2], Randall William Davis[3], Natsuki Konno[1], Kyohei Murayama[1], Satsuki Ochiai[1], Yoko Mitani[4]

1 Graduate School of Environmental Science, Hokkaido University, Hakodate, Hokkaido, Japan, 2 Akkeshi Marine Station, Field Science Center for Northern Biosphere, Hokkaido University, Akkeshi, Hokkaido, Japan, 3 Department of Marine Biology, Texas A&M University at Galveston, Galveston, Texas, United States of America, 4 Wildlife Research Center, Kyoto University, Kyoto, Japan

☯ These authors contributed equally to this work.
* jackson.johnstone.m5@elms.hokudai.ac.jp

**Data Availability Statement:** All relevant data are within the manuscript and its Supporting

## Abstract

In recent decades, the locally extinct sea otter (*Enhydra lutris lutris*) has been recolonizing the coast of eastern Hokkaido. Their diet includes benthic invertebrates such as bivalves, sea urchins, snails, and chitons. In the fall of 2021, a harmful algal bloom (HAB) of *Karenia selliformis* occurred across Hokkaido's northern and eastern coasts, leading to a massive mortality of sea urchins. This dinoflagellate produces a neurotoxin (gymnodimine) implicated in shellfish poisoning. To determine the effect of the HAB on the marine community, we conducted benthic surveys using SCUBA and visually monitored the prey items of the sea otters in the affected area from 2020 to 2023. Following the HAB, we observed an 82% decrease in benthic sea urchin density (number $m^2$), leading to their complete absence from the diet of sea otters. Conversely, bivalve density increased six-fold, accompanied by a nearly two-fold rise in their percentage in the sea otters' diet. Minimal changes were observed in the density of chitons and snails, with no significant alteration in the sea otters' diet. Despite these changes, the impact of the HAB on otters' dietary preferences was temporary, as the percentage of dietary sea urchins began recovering one year later. Sea otters augmented their diet with bivalves to compensate for the reduced availability of sea urchins during the HAB with no apparent effects on the number of sea otters or their health. Our results highlight the adaptability of sea otters to adjust their diet according to prey availability.

## Introduction

Red tides, or harmful algal blooms (HABs), are generated by the proliferation of specific phytoplankton varieties, such as dinoflagellates or diatoms. They result in a reddish or brownish

Information files. If any additional data is required, we will be happy to provide it as well.

**Funding:** This study was funded by the Asahi Glass Foundation (https://www.af-info.or.jp/en/research/) in 2020-2021 (to I.S. and Y.M.), the Sasakawa Peace Foundation (https://www.spf.org/en/) in 2021 (to S.O. and Y.M.), the Hokkaido University DX Doctoral Fellowship (https://sites.google.com/eis.hokudai.ac.jp/dxphd-fellow/home) for 2022-2023 (to J.J.), the Pro Natura Fund (https://www.pronaturajapan.com/en/index.html) (to I.S., R.D., and Y.M.), the JSPS (Japanese Society for the Promotion of Science) Invitational Fellowships for Research in Japan (https://www.jsps.go.jp/english/e-inv/) (to R.D. and Y.M.), and the Environment Research and Technology Development Fund (JPMEERF20234003) of the Environmental Restoration and Conservation Agency provided by the Ministry of the Environment of Japan to IS. The funders had no role in study design, data collection, and analysis, the decision to publish, or the preparation of the manuscript.

**Competing interests:** The authors have declared that no competing interests exist.

tint to the water and produce toxins that can lead to extensive mortality among marine organisms, including fish and shellfish. Warm water dinoflagellates, such as *Karenia brevis*, can have long-term effects on ecosystems and the economies of local fisheries. *K. brevis* events have been linked to the mortality of fish and invertebrates in the Gulf of Mexico for over 50 years. The west coast of Florida has experienced several HABs, resulting in the mortality of various marine species. Higher-trophic organisms, such as dolphins and manatees, are also vulnerable to the effects of these toxins. Consequently, HABs have been extensively investigated over the past several decades, often serving as the focus of national and international environmental policy discussions. Commercial and artisanal fisheries rank among the most crucial industries in the Japanese economy, directly or indirectly impacting millions of Japanese citizens. HABs can profoundly disrupt local fisheries, resulting in considerable economic losses. In severe instances, such events may necessitate the complete closure of fisheries in affected areas.

HABs have been documented in Japan for centuries, with records dating back to 731 CE [1]. Many of these occurrences have been observed in southern Japan, particularly in regions like the Seto Inland Sea [2]. The heightened frequency of HABs over the last 50 years, extending to higher latitudes, correlates with increasing global temperatures [3, 4]. Historically, open ocean HABs have been rare in the cold waters around Hokkaido, with occurrences primarily concentrated in shallow coastal waters and bays. [5, 6]. However, in 2021, a HAB linked to ocean temperatures 1–3˚C above average occurred in eastern Hokkaido, resulting in extensive fish and sea urchin mortality [5, 7, 8]. This event significantly affected the sea urchin fishery, which relies on hatchery production for reseeding coastal habitats [5, 9, 10].

We used the HAB in 2021 to study the ecosystem-level effects in eastern Hokkaido. Sea otters in this area were extirpated during the 19th-century Maritime Fur Trade but began recolonizing the area in the 1980s [11–13]. While there is limited information regarding the diet of sea otters in Hokkaido, we have observed them eating bivalves (*Clinocardium californiense*, *Callista brevisiphonata*, and others), sea urchins (*Strongylocentrotus intermedius*), crabs (*Paralithodes brevipes* and *Telmessus cheiragonus*), snails, and chitons (*Cryptochiton stelleri*) [14]. Bivalves are the predominant prey for sea otters in our study area, but sea urchins and crabs are also important. In response to the HAB, our null hypothesis was that sea otters would accommodate to the reduced abundance of sea urchins by shifting to other prey, with minimal effects on health and behavior. The alternate hypothesis was that the HAB would severely affect sea otters, including decreased food consumption, emaciation, and possibly death. We also hypothesized that any dietary shifts would depend on the recovery of benthic invertebrates, especially sea urchins. The objective of this paper was to examine the impact of a red tide on the feeding behaviors of sea otters and their associated environments. The sea otter population's response to the red tide provided a unique opportunity to observe the effects of a significant environmental event on sea otters in Hokkaido for the first time. Additionally, this research encompasses the first comprehensive and long-term observation of feeding behavior in this newly recolonizing sea otter population in Hokkaido. The study addresses critical gaps in knowledge regarding this expanding population, which is expected to become increasingly relevant as it continues to grow and interact more frequently with human activities and the fisheries industry.

## Materials and methods

### Study area

Our study area was the Moyururi and Yururi Islands (43.2219˚, 145.6244˚), about 4 km from the coast and 12 km from Nemuro in Eastern Hokkaido (Fig 1). This area has commercial fisheries for sea urchins, crabs, and various species of kelp.

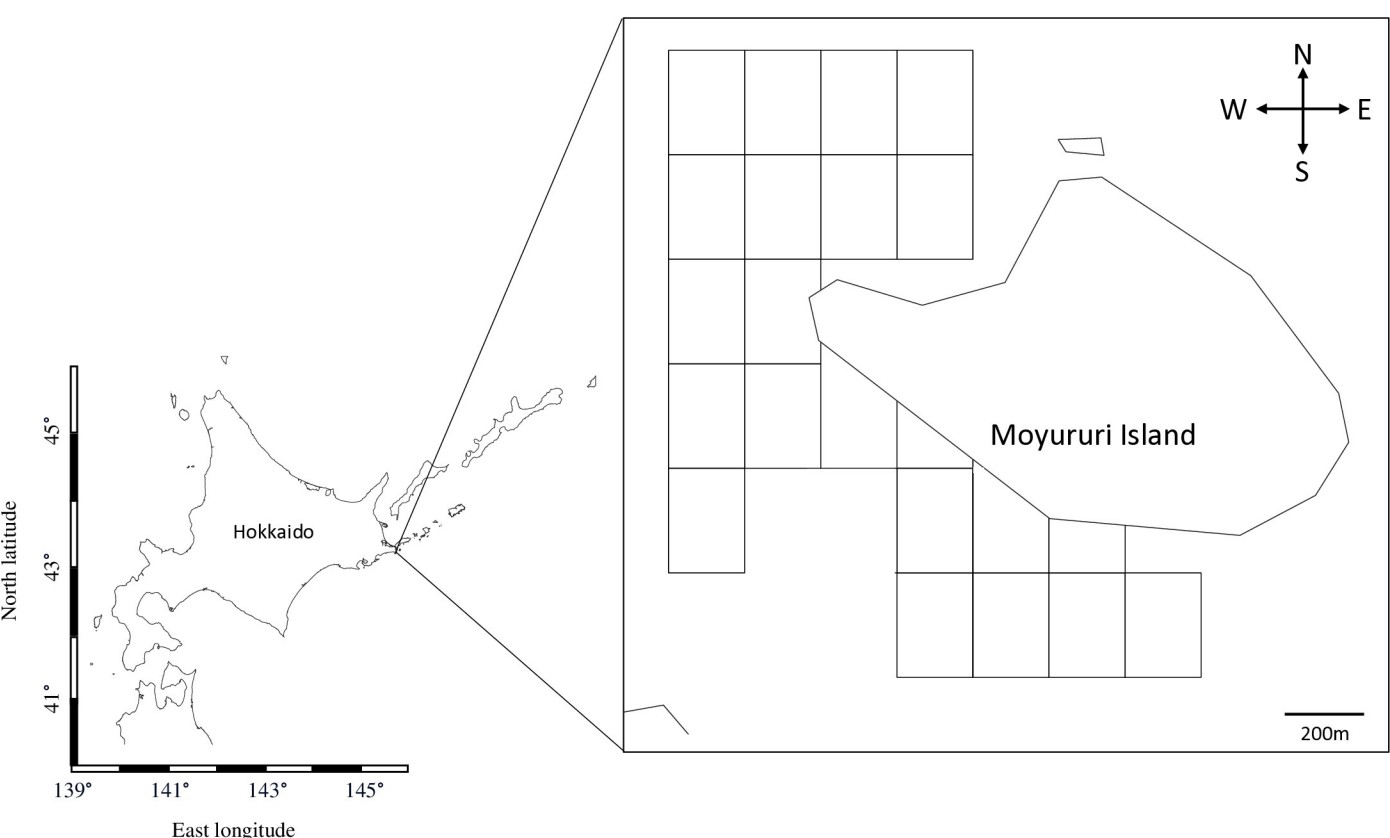

**Fig 1. The study area for the benthic surveys at Moyururi Island in Eastern Hokkaido.** Square blocks show the 200 m$^2$ quadrats surveyed utilizing random quadrat sampling.

## Benthic survey

Benthic surveys were conducted using SCUBA in September 2020, 2022, and 2023 in an area of concentrated sea otter foraging activity along the west coast of Moyururi Island. The sea area was divided by twenty 200 m x 200 m grids, and each grid was designated one point. Quadrats used were 50 cm x 50 cm size, as were used in Kvitek et al. (1993) [15], and we deployed them randomly four times at each point. Random placement of quadrats, and collection of benthic invertebrates within was conducted by a hired professional diver. Crabs were counted but not collected, so morphometrics (length, width, and mass) were not recorded. They are also significantly more mobile than other benthic organisms collected, so it would be difficult to confirm the number of crabs in the survey area. All other specimens were frozen at -20°C until species identification and morphometrics were recorded later. Specimens were divided into three size classes: small (2–5 cm), medium (5–10 cm), and large (>10 cm), following Kvitek et al. (1993) [15]. The number of specimens counted or collected was used to estimate each species' average seafloor density (number m$^{-2}$).

## Sea otter diet

We conducted observations of sea otter foraging behavior using binoculars (Nikon StabilEyes 16x32, Nikon, Minato City, Tokyo, Japan) during 30-minute focal follows from June to August 2020, June to September 2021, and May to September 2022 from a small boat. The location of each focal follow was recorded using GPS (Garmin, Olathe, Kansas, USA). When a sea otter

began feeding, we documented dive duration, prey type (to the highest specificity possible), prey number, prey size, and inter-dive interval. Only focal follows with 5–10 foraging dives were included in our analysis to avoid sampling bias [16, 17]. Prey size was estimated relative to the width of a sea otter paw (i.e., 5 cm; [15]).

## Data analysis

We compared the seafloor densities for bivalves, sea urchins, snails, chitons, and crabs pre-(2020) and post-HAB (2022, 2023). Additionally, we compared sea otter prey compositions (percentage of dietary bivalves, crabs, sea urchins, chitons, and snails) before (2020–2021), immediately after (2022), and one year after (2023) the HAB. Prey species that could not be identified were categorized as unknown and excluded from the analysis. Other prey items that were anecdotally observed, such as sea cucumbers and hermit crabs, were also excluded from the analysis. A Shapiro-Wilks normality test confirmed normal distributions for both benthic organism number and sea otter dietary percentage data. Tukey's HSP tests were conducted to identify differences [17–21].

## Ethics statement

We received permission from the local fisheries authorities to collect marine invertebrates for our research study. All research conducted since 2022 was per section II (the effects of observation and the impact on habitat) of the Guidelines for Wildlife Research issued by the Wildlife Research Center of Kyoto University (Permit Number WRCY-2022-022). Prior to 2022, Hokkaido University did not require any such ethics statement, as all research conducted around sea otters was non-invasive.

## Results

### Benthic surveys

On average, 235 benthic specimens were collected from 20 grids (Table 1) during each survey. Before the HAB (2020), sea urchins (6.7 per $m^2$) had the highest seafloor density, followed by snails (1.5 per $m^2$), chitons (0.6 per $m^2$), and bivalves (0.3 per $m^2$). Immediately after the HAB (2022), sea urchins (1.2 per $m^2$) decreased significantly (82%) from pre-HAB levels (Tukey's HSD test, $p < 0.05$), while bivalves (1.8 per $m^2$) displayed a 6-fold increase ($p = 0.066$) (Fig 2). No significant differences were observed for chitons (0.4 per $m^2$) and snails (2.8 per $m^2$) between 2020 and 2022 (Chitons $p = 0.845$, Snails $p = 0.364$). One year after the HAB (2023), the density of bivalves (1.3 per $m^2$, $p = 0.712$) declined 28% from levels immediately after the HAB but remained elevated by 4.3-fold compared to pre-HAB densities. Likewise, snails (2.0 per $m^2$, $p = 0.645$) decreased by 29%, sea urchins (2.3 per $m^2$, $p = 0.115$) increased by 2-fold, and chitons (0.5 per $m^2$, $p = 0.959$) remained unchanged.

**Table 1. Seafloor densities (per $m^2$) of invertebrates sampled during surveys before (2020) and after (2022 and 2023) the HAB.**

|  | Density (number $m^2$) | | | | Total number collected |
|---|---|---|---|---|---|
|  | Urchin | Chiton | Bivalves | Snail |  |
| **Pre-HAB** |  |  |  |  |  |
| 2020 | 6.7 | 0.6 | 0.3 | 1.5 | 257 |
| **Post-HAB** |  |  |  |  |  |
| 2022 | 1.2 | 0.4 | 1.8 | 2.8 | 230 |
| 2023 | 2.3 | 0.5 | 1.3 | 2.0 | 219 |

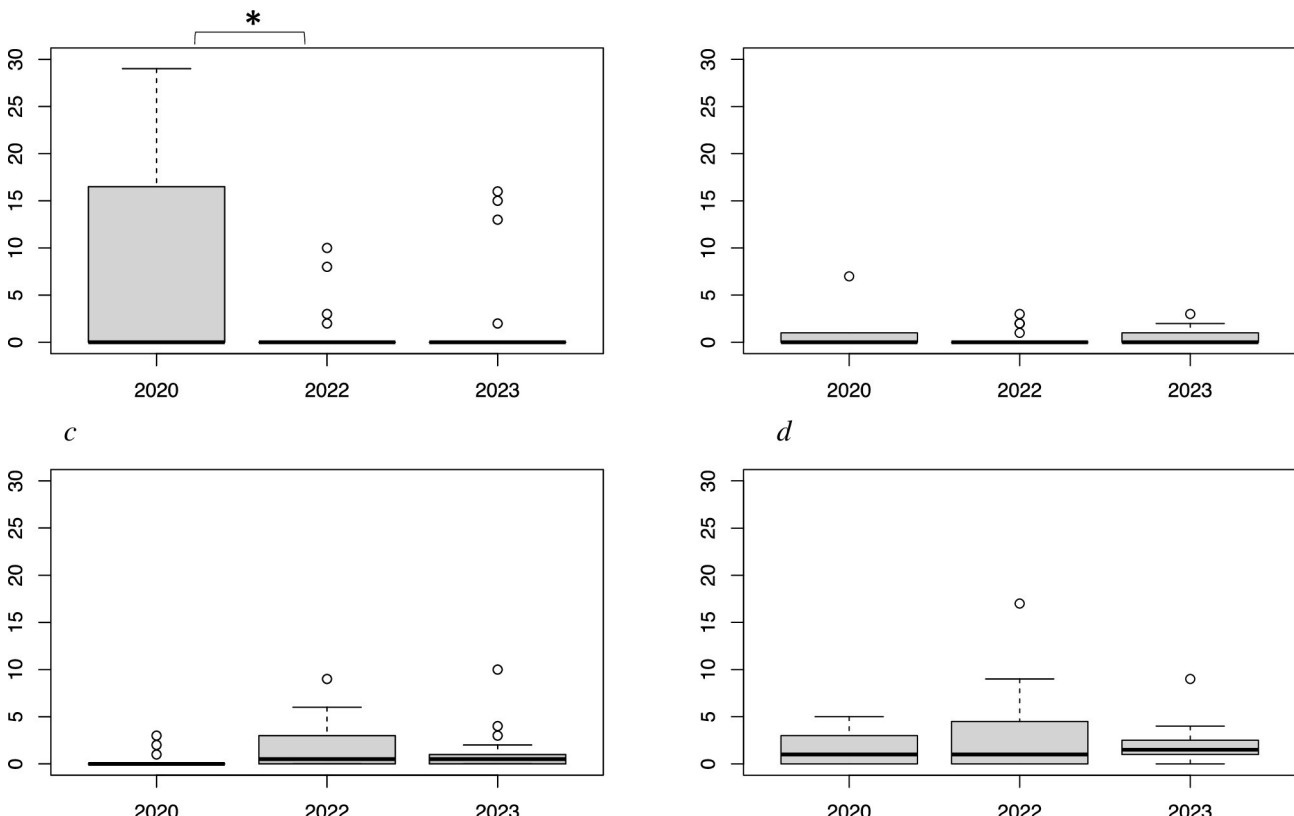

**Fig 2.** Number (per m²) of major sea otter prey items sampled from SCUBA survey: *a* (Sea Urchins), *b* (Chitons), *c* (Bivalves), *d* (Snails) retrieved during benthic quadrat surveys of 2020, 2022, and 2023. Asterisks (*) above the graph display a significant difference (p < 0.05) between two years.

## Sea otter diet

From 2020–2022, 265 foraging dives during 34 focal follows were monitored, and 508 prey items were recorded (Table 2). Before the HAB (October 2021), there were no significant differences in the percentages of prey captured during focal follows (Table 2). The predominate prey were bivalves ($\bar{x}$ = 34.8%; range 31.8–36.5%), crabs ($\bar{x}$ = 13.7%; range 12.9–15.3%), and sea urchins ($\bar{x}$ = 7.8; range 4.7–13.7%). The average percentages for chitons and snails were less than 5%. Immediately after the HAB in 2022, the percentage of dietary bivalves increased significantly (2-fold) to 67.3% (Tukey's HSD test, p < 0.05), while sea urchins completely disappeared from the diet (Fig 3). One year after the HAB (2023), the percentage of sea urchins increased significantly to 13.9% (p < 0.05) and the percentage of bivalves returned to near the pre-HAB level of 37.5% (p < 0.05). The percentages of chitons, snails, and crabs in the diet were not significantly different across the entire survey period.

## Discussion

Our benthic surveys revealed an 82% decrease in sea urchin density and a 6-fold increase in the number of bivalves immediately after the HAB in 2022 (Table 1). By the following year (2023), the average sea urchin density showed partial recovery, while clam density remained elevated compared to 2020. However, our surveys were not comprehensive and susceptible to interannual sampling bias. Nevertheless, our data indicated a pronounced decrease in sea

**Table 2. Mean (+1 SD) percentage of prey in focal follow pre-HAB (2020, 2021) and post-HAB (2022 and 2023).**

|  | Urchin | Chiton | Bivalve | Snail | Crab | Unknown | Bouts | Successful Dives |
|---|---|---|---|---|---|---|---|---|
| **Pre-HAB** |  |  |  |  |  |  |  |  |
| 2020/2021 | 7.9 + 14.5 | 3.8 + 6.4 | 34.9 + 34.4 | 4.5 + 9.4 | 13.8 + 27.5 | 35.2 + 32.9 | 20 | 150 |
| **Post-HAB** |  |  |  |  |  |  |  |  |
| 2022 | 0 + 0 | 2.0 + 3.4 | 67.3 + 34.2 | 7.0 + 26.1 | 10.7 + 23.5 | 12.6 + 22.8 | 14 | 115 |
| 2023 | 13.9 + 19.7 | 1.5 + 6.3 | 37.5 + 37.0 | 8.7 + 18.4 | 11.8 + 21.1 | 25.4 + 23.1 | 24 | 201 |

urchin density after the HAB, with a partial recovery one year later. Additionally, clam density appeared to increase post-HAB and remain elevated.

An earlier study showed that the HAB caused mass mortality of sea urchins and moderate effects on chitons and snails throughout the coastal waters of eastern Hokkaido, similar to HABs in other regions [2, 5, 22–25]. Consequently, the HAB likely caused the decrease in sea urchin density and distribution around the Moyururi and Yururi Islands. Reseeding with hatchery-raised sea urchins did not occur in our study area after the HAB, so the partial recovery represents a natural process that may take about three years before sea urchins reach the size consumed by sea otters (Ochiishii Fisheries Cooperative, pers. comm.).

We observed an increase in the density of bivalves after the HAB. Bivalves filter phytoplankton from the water column for nutrition and may exhibit higher growth rates during HABs [26]. Certain algal blooms, such as brown tide algae (*Aureococcus anophagefferens*) along the northeastern coast of the United States, can adversely affect bivalves by inhibiting larval growth [27, 28]. However, some bivalves are not affected by the brevetoxin-producing algae *Karenia selliformis* [29]. Generally, more intense planktonic blooms are associated with higher water temperatures [30, 31], and such conditions were observed before the HAB in our study

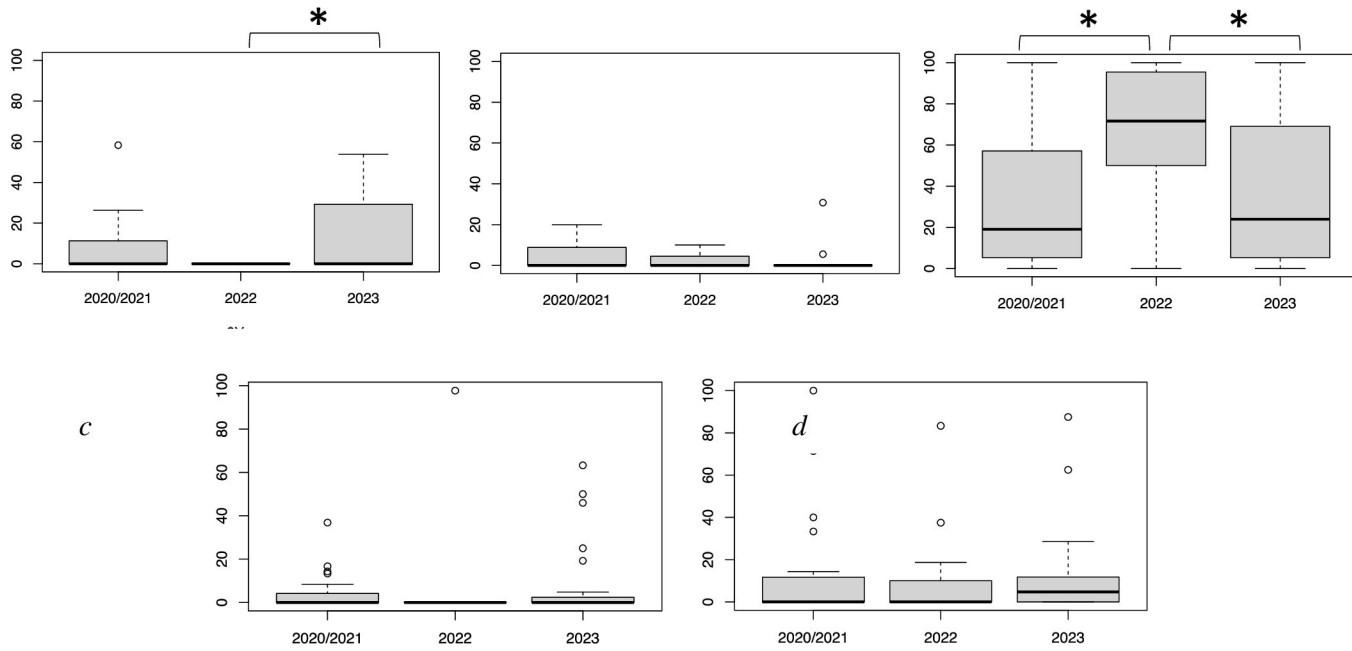

**Fig 3.** Observed average proportion of major prey items of sea otters: *a* (Sea Urchins), *b* (Chitons), *c* (Bivalves), *d* (Snails), and *e* (Crabs) per bout in the Sea Otter diets at Moyururi and Yururi Islands from 2020/ 2021, 2022, and 2023. *Error bars* represent +1 SE. Asterisks (*) above the graph display a significant difference (p < 0.05) between two or more years.

area [8]. Our results may indicate enhanced bivalve growth, as small (2–5 cm) bivalves significantly increased abundance (Tukey's HSD Test, $p < 0.05$). However, we cannot rule out inter-annual sampling bias. Chitons and snails did not significantly change in abundance over the three years.

We observed a shift in the sea otter diet immediately after the HAB and during the recovery period one year later. Before the HAB, sea urchins accounted for 9.2% (range 4.7–13.7%) of the diet. However, following the HAB, sea urchins disappeared from the diet (Table 2), reflecting their diminished abundance. Despite this, the number of sea otters in the study area remained constant (Suzuki et al., in prep). Bivalves were the predominant prey (34.1%) before the HAB, with their contribution to the diet doubling to 67.3% after the HAB (Table 2). The likely cause for this increase was prey-switching when sea urchins decreased in abundance. This switch may have been facilitated by the enhanced growth of bivalves, leading to increased abundance in larger size classes. Our results suggest that as generalists [32], sea otters responded to the shifting benthic community structure by consuming prey that became more abundant because of the HAB.

Sea otter populations in California and the Aleutian Islands, Alaska, show a greater preference for sea urchins compared with those in eastern Hokkaido [16, 17, 33, 34]. Sea otter dietary preference correlates with prey abundance, and they do not necessarily require a wide range of prey to thrive [35–38]. The unusual HAB [39, 40] in eastern Hokkaido provided an opportunity to observe the ecosystem effects at two trophic levels [5, 7, 9, 10]. Because of the dietary adaptability of sea otters [32], the HAB had no obvious effect on the health and abundance of sea otters in our study area. As sea urchins recover, their dietary contribution may return to pre-HAB levels, especially with the reseeding of small, hatchery sea urchins.

## Supporting information

**S1 Appendix.** The total number of small, medium, and large bivalves (a) collected during SCUBA dive surveys conducted in September 2020, 2022, and 2023. Number (per $m^2$) of b (small), c (medium), and d (large) bivalves retrieved during benthic quadrat surveys of 2020, 2022 and 2023. Asterisks (*) above the graph display a significant difference ($p < 0.05$) between two years.
(DOCX)

**S1 File.**
(PDF)

**S2 File.**
(DOCX)

## Acknowledgments

We thank Professors K. Miyashita and M. Nakaoka from Hokkaido University Fisheries Science Center for providing advice and access to research facilities. Captain Y. Kotani supported sea otter observations and piloted the research vessel, and T. Tani conducted the benthic surveys. We thank students for their assistance in the benthic specimen analysis. Hokkaido University allowed the use of the Hakodate Fisheries Science Center and the Akkeshi Marine Station for research and lodging. Special thanks to the Ochiishi and Nemuro Fisheries Cooperatives for permitting the benthic sampling and information on the reseeding of hatchery-raised sea urchins.

## Author Contributions

**Conceptualization:** Jackson Johnstone, Ippei Suzuki, Yoko Mitani.

**Funding acquisition:** Jackson Johnstone, Ippei Suzuki, Randall William Davis, Yoko Mitani.

**Investigation:** Jackson Johnstone, Ippei Suzuki, Natsuki Konno, Kyohei Murayama, Satsuki Ochiai, Yoko Mitani.

**Methodology:** Jackson Johnstone, Ippei Suzuki, Randall William Davis, Satsuki Ochiai, Yoko Mitani.

**Project administration:** Ippei Suzuki, Yoko Mitani.

**Visualization:** Jackson Johnstone, Yoko Mitani.

**Writing – original draft:** Jackson Johnstone, Ippei Suzuki.

**Writing – review & editing:** Jackson Johnstone, Ippei Suzuki, Randall William Davis, Natsuki Konno, Kyohei Murayama, Satsuki Ochiai, Yoko Mitani.

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
