## [Decision Letter · Decision Letter 0]

31 Jul 2024

PONE-D-24-15918The effect of a harmful algal bloom (Karenia selliformis) on the benthic invertebrate community and the sea otter (Enhydra lutris) diet in eastern Hokkaido.PLOS ONE

Dear Dr. Johnstone,

Thank you for submitting your manuscript to PLOS ONE. After careful consideration, we feel that it has merit but does not fully meet PLOS ONE’s publication criteria as it currently stands. Therefore, we invite you to submit a revised version of the manuscript that addresses the points raised during the review process.

We look forward to receiving your revised manuscript.

Kind regards,

Arumugam Sundaramanickam, PhD

Academic Editor

PLOS ONE

Journal Requirements:

"This study was funded by the Asahi Glass Foundation (https://www.af-info.or.jp/en/research/) in 2020-2021 (to I.S. and Y.M.), the Sasakawa Peace Foundation (https://www.spf.org/en/) in 2021 (to S.O. and Y.M.), the Hokkaido University DX Doctoral Fellowship (https://sites.google.com/eis.hokudai.ac.jp/dxphd-fellow/home) for 2022-2023 (to J.J.), the Pro Natura Fund (https://www.pronaturajapan.com/en/index.html) (to I.S., R.D., and Y.M.), and the JSPS (Japanese Society for the Promotion of Science) Invitational Fellowships for Research in Japan (https://www.jsps.go.jp/english/e-inv/) (to R.D. and Y.M.)."

"This study was funded by the Asahi Glass Foundation in 2020-2021 (to I.S. and Y.M.), the Sasakawa Peace Foundation in 2021 (to S.O. and Y.M.), the Hokkaido University DX Doctoral Fellowship for 2022-2023 (to J.J.), the Pro Natura Fund (to I.S., R.D., and Y.M.), and the JSPS (Japanese Society for the Promotion of Science) Invitational Fellowships for Research in Japan (to R.D. and Y.M.)."

"This study was funded by the Asahi Glass Foundation (https://www.af-info.or.jp/en/research/) in 2020-2021 (to I.S. and Y.M.), the Sasakawa Peace Foundation (https://www.spf.org/en/) in 2021 (to S.O. and Y.M.), the Hokkaido University DX Doctoral Fellowship (https://sites.google.com/eis.hokudai.ac.jp/dxphd-fellow/home) for 2022-2023 (to J.J.), the Pro Natura Fund (https://www.pronaturajapan.com/en/index.html) (to I.S., R.D., and Y.M.), and the JSPS (Japanese Society for the Promotion of Science) Invitational Fellowships for Research in Japan (https://www.jsps.go.jp/english/e-inv/) (to R.D. and Y.M.)."

5. Please ensure that you refer to Figure 1 in your text as, if accepted, production will need this reference to link the reader to the figure.

6. Please include a separate caption for each figure in your manuscript.

7. Please include your tables as part of your main manuscript and remove the individual files. Please note that supplementary tables (should remain/ be uploaded) as separate ""supporting information"" files

Reviewers' comments:

Reviewer's Responses to Questions

**Comments to the Author**

1. Is the manuscript technically sound, and do the data support the conclusions?

Reviewer #1: Yes

2. Has the statistical analysis been performed appropriately and rigorously? 

Reviewer #1: Yes

3. Have the authors made all data underlying the findings in their manuscript fully available?

Reviewer #1: Yes

4. Is the manuscript presented in an intelligible fashion and written in standard English?

Reviewer #1: Yes

5. Review Comments to the Author

Reviewer #1: The authors are to be commended for the work presented in this manuscript. Information about the biology and natural history of Hokkaido's sea otter population is scarce, so any additions are important assets to sea otter conservation. Overall, I feel that this paper is well written, contains important information, and I only have a few suggestion for minor edits.

The overall purpose of this study and subsequent manuscript is not clearly stated. I recommend that the authors revisit how the purpose is described to ensure that it is clear to the reader.

Line 100-111: I believe that the benthic survey methodology is valid, but the methods section would be strengthened by including reference to similar efforts that have already been through review and published.

Lines 41-42 & 208-209: The authors should expand their discussion on how they came to the conclusion that there were no health effects on the otters.

6. PLOS authors have the option to publish the peer review history of their article (what does this mean?). If published, this will include your full peer review and any attached files.

Reviewer #1: No

---

## [Author Response · Author response to Decision Letter 0]

20 Sep 2024

Response to editor revisions: 

-Thank you very much for your comments. I have reformatted the titles for the documents in my submission.

"This study was funded by the Asahi Glass Foundation (https://www.af-info.or.jp/en/research/) in 2020-2021 (to I.S. and Y.M.), the Sasakawa Peace Foundation (https://www.spf.org/en/) in 2021 (to S.O. and Y.M.), the Hokkaido University DX Doctoral Fellowship (https://sites.google.com/eis.hokudai.ac.jp/dxphd-fellow/home) for 2022-2023 (to J.J.), the Pro Natura Fund (https://www.pronaturajapan.com/en/index.html) (to I.S., R.D., and Y.M.), and the JSPS (Japanese Society for the Promotion of Science) Invitational Fellowships for Research in Japan (https://www.jsps.go.jp/english/e-inv/) (to R.D. and Y.M.)."

-I have added a section that includes the lack of a role for the funders from the study construction in the Cover letter.

"This study was funded by the Asahi Glass Foundation in 2020-2021 (to I.S. and Y.M.), the Sasakawa Peace Foundation in 2021 (to S.O. and Y.M.), the Hokkaido University DX Doctoral Fellowship for 2022-2023 (to J.J.), the Pro Natura Fund (to I.S., R.D., and Y.M.), and the JSPS (Japanese Society for the Promotion of Science) Invitational Fellowships for Research in Japan (to R.D. and Y.M.)."

"This study was funded by the Asahi Glass Foundation (https://www.af-info.or.jp/en/research/) in 2020-2021 (to I.S. and Y.M.), the Sasakawa Peace Foundation (https://www.spf.org/en/) in 2021 (to S.O. and Y.M.), the Hokkaido University DX Doctoral Fellowship (https://sites.google.com/eis.hokudai.ac.jp/dxphd-fellow/home) for 2022-2023 (to J.J.), the Pro Natura Fund (https://www.pronaturajapan.com/en/index.html) (to I.S., R.D., and Y.M.), and the JSPS (Japanese Society for the Promotion of Science) Invitational Fellowships for Research in Japan (https://www.jsps.go.jp/english/e-inv/) (to R.D. and Y.M.)."

-I have removed this statement from the acknowledgments section of my manuscript and would like to include the following at the bottom of a revised cover letter file: 

Funding: This study was funded by the Asahi Glass Foundation (https://www.af-info.or.jp/en/research/) in 2020-2021 (to I.S. and Y.M.), the Sasakawa Peace Foundation (https://www.spf.org/en/) in 2021 (to S.O. and Y.M.), the Hokkaido University DX Doctoral Fellowship (https://sites.google.com/eis.hokudai.ac.jp/dxphd-fellow/home) for 2022-2023 (to J.J.), the Pro Natura Fund (https://www.pronaturajapan.com/en/index.html) (to I.S., R.D., and Y.M.), and the JSPS (Japanese Society for the Promotion of Science) Invitational Fellowships for Research in Japan (https://www.jsps.go.jp/english/e-inv/) (to R.D. and Y.M.).

Role of funder: The funders had no role in study design, data collection, and analysis, the decision to publish, or the preparation of the manuscript.

-I have included the ethics statement as point 5 of my materials and methods section of the manuscript. I have also included the code of conduct established by the Kyoto University Wildlife Research Center that we followed during the course of our research (168-176).

5. Please ensure that you refer to Figure 1 in your text as, if accepted, production will need this reference to link the reader to the figure.

-I have added a reference within text to figure 1 in the study area point of the materials and methods section of the revised manuscript (Line 109).

6. Please include a separate caption for each figure in your manuscript.

-Each figure has a title/ caption

7. Please include your tables as part of your main manuscript and remove the individual files. Please note that supplementary tables (should remain/ be uploaded) as separate ""supporting information"" files

-I have included the tables within the revised manuscript, in lines 191-194 and 208-211. 

-Supporting information caption is now included at the end of the manuscript.

Reviewer’s comments section:

Reviewer #1: The authors are to be commended for the work presented in this manuscript. Information about the biology and natural history of Hokkaido's sea otter population is scarce, so any additions are important assets to sea otter conservation. Overall, I feel that this paper is well written, contains important information, and I only have a few suggestion for minor edits.

-Thank you very much for your kind words and comments and suggestions.

The overall purpose of this study and subsequent manuscript is not clearly stated. I recommend that the authors revisit how the purpose is described to ensure that it is clear to the reader.

-I have tried to add more clarity to the purpose of the experiment towards the end of my introduction section, lines 97-105. I reemphasized the novelty of this research, in how it offers one of the first opportunities to observe this population and a unique opportunity to observe the effects of a red tide upon it. I also directly stated what our research seeks to do (our purpose) to go with what we already stated were the goals of the research.

Line 100-111: I believe that the benthic survey methodology is valid, but the methods section would be strengthened by including reference to similar efforts that have already been through review and published.

-The survey methodology is taken from the same paper (Kvitek et al., 1993) that we followed for designing the protocol for deciding benthic organism side classes. I have included another reference to the paper on line 116. The Kvitek paper is foundational to the field of sea otter dietary preference in regards to the availability of prey items.

Lines 41-42 & 208-209: The authors should expand their discussion on how they came to the conclusion that there were no health effects on the otters.

-The existing line about the population remaining unchanged provides the evidence we have that there were no observed negative health effects on the population. I am using the yet-unpublished research data (Suzuki et al., in prep) as evidence for this lack

---

## [Decision Letter · Decision Letter 1]

6 Nov 2024

The effect of a harmful algal bloom (Karenia selliformis) on the benthic invertebrate community and the sea otter (Enhydra lutris) diet in eastern Hokkaido.

PONE-D-24-15918R1

Dear Dr. Johnstone,

We’re pleased to inform you that your manuscript has been judged scientifically suitable for publication and will be formally accepted for publication once it meets all outstanding technical requirements.

Kind regards,

Arumugam Sundaramanickam, PhD

Academic Editor

PLOS ONE

Additional Editor Comments (optional):

Reviewers' comments:

Reviewer's Responses to Questions

**Comments to the Author**

1. If the authors have adequately addressed your comments raised in a previous round of review and you feel that this manuscript is now acceptable for publication, you may indicate that here to bypass the “Comments to the Author” section, enter your conflict of interest statement in the “Confidential to Editor” section, and submit your "Accept" recommendation.

Reviewer #1: All comments have been addressed

2. Is the manuscript technically sound, and do the data support the conclusions?

Reviewer #1: Yes

3. Has the statistical analysis been performed appropriately and rigorously? 

Reviewer #1: Yes

4. Have the authors made all data underlying the findings in their manuscript fully available?

Reviewer #1: Yes

5. Is the manuscript presented in an intelligible fashion and written in standard English?

Reviewer #1: Yes

6. Review Comments to the Author

Reviewer #1: Thank you for your hard work in this challenging area and with a challenging marine mammal. You have adequately addressed my expressed concerns.

7. PLOS authors have the option to publish the peer review history of their article (what does this mean?). If published, this will include your full peer review and any attached files.

Reviewer #1: No

---

## [Editor Report · Acceptance letter]

11 Nov 2024

PONE-D-24-15918R1 

PLOS ONE

Dear Dr. Johnstone, 

I'm pleased to inform you that your manuscript has been deemed suitable for publication in PLOS ONE. Congratulations! Your manuscript is now being handed over to our production team.

Kind regards, 

on behalf of

Professor Arumugam Sundaramanickam 

Academic Editor

PLOS ONE